# Highly Sensitive Detection of miR-200c in Metastatic Lymph Nodes Using Scanning Single-Molecule Counting

**DOI:** 10.3390/cancers17193133

**Published:** 2025-09-26

**Authors:** Yuki Sata, Terunaga Inage, Takahiro Nakajima, Yuki Ito, Ichiro Yoshino, Hidemi Suzuki

**Affiliations:** 1Department of General Thoracic Surgery, Graduate School of Medicine, Chiba University, Chiba 260-8670, Japan; 2Department of General Thoracic Surgery, Dokkyo Medical University, Mibu, Shimotsuga, Tochigi 321-0293, Japan

**Keywords:** molecular nodal staging, miRNA, scanning single-molecule counting (SSMC), endobronchial ultrasound-guided transbronchial needle aspiration, lymph node metastasis

## Abstract

The novel 9th TNM classification of lung cancer distinguishes between single-station N2 (N2a) and multi-station N2 (N2b) disease, refining nodal staging. Endobronchial ultrasound-guided transbronchial needle aspiration (EBUS-TBNA) is a highly diagnostic, minimally invasive tool for this purpose, with rapid on-site evaluation recommended for improved efficiency. To reduce false-negative diagnoses, we explored microRNA biomarkers as a complementary approach to EBUS-TBNA. This study assessed the feasibility and clinical utility of scanning single-molecule counting (SSMC) for diagnostic samples, offering faster and more convenient miRNA detection than conventional RT-qPCR. We evaluated SSMC’s ability to detect miR-200c in metastatic lymph nodes, demonstrating its potential for highly sensitive molecular nodal staging.

## 1. Introduction

Lung cancer is the leading cause of cancer-related deaths, with 1.7 million deaths per year [1]. Treatment strategies for lung cancer depend on its stage and comprise different approaches, including surgery, radiotherapy, chemotherapy, targeted therapy, immunotherapy, and multimodal treatment. For N2 disease, a novel neoadjuvant strategy combining chemotherapy and immune checkpoint inhibitors has shown promising results in locally advanced disease [2,3]. This approach achieved a pathological complete response exceeding 30% (no residual disease within surgically resected materials), indicating the importance of obtaining preoperative diagnostic samples for additional ancillary testing. Furthermore, the novel 9th TNM classification of lung cancer differentiates N2 disease into single-station N2 (N2a) and multi-station N2 (N2b), which affects staging [4,5]. For example, T1N2a disease was classified as stage IIIA in the 8th TNM staging system but is now categorized as stage IIB in the 9th TNM. Therefore, a more accurate and efficient N2 diagnosis is critical for determining the treatment method.

Endobronchial ultrasound-guided transbronchial needle aspiration (EBUS-TBNA) is a well-established, minimally invasive tool for sampling mediastinal and hilar lymph nodes [6]. Although EBUS-TBNA has high sensitivity and specificity for diagnosing lymph node metastasis, discrepancies may still occur between preoperative and final pathological staging due to nodal migration [7,8]. The limited negative predictive value (NPV) of needle biopsies is often discussed [9,10]. Therefore, current guidelines recommend additional surgical staging in cases with negative needle biopsy results when the clinical suspicion of metastasis is high [6]. However, surgical staging is more complicated than EBUS-TBNA. To reduce false-negative diagnoses, we are developing biomarkers to complement EBUS-TBNA. In our previous studies, we focused on microRNAs (miRNAs), particularly miR-200c [11,12]. miRNAs are a large group of small, non-coding RNAs that regulate the post-transcriptional expression of target genes and are highly stable, even under conditions that completely degrade mRNA and in formalin-fixed paraffin-embedded samples [13,14,15].

Furthermore, we simultaneously assessed the feasibility and clinical utility of scanning single-molecule counting (SSMC) for diagnostic samples (Figure 1). The SSMC system uses confocal microscopy to count each fluorescent molecule in a solution while scanning the observation region. Using the SSMC system, results can be obtained in approximately 75 min (standard method), compared to 2–3 h with conventional quantitative real-time polymerase chain reaction (RT-qPCR), because SSMC evaluates miRNAs directly without reverse transcription or PCR amplification. This enables faster and more convenient measurements than conventional RT-qPCR [16,17].

The schematic illustrates the workflow of the SSMC system. Fluorescently labeled probes hybridize directly to target miRNAs in the sample solution. Using confocal laser scanning microscopy, the observation volume is illuminated, and individual fluorescent signals corresponding to single molecules are detected by highly sensitive photodetectors. Each detected fluorescent event is counted as one molecule, enabling absolute quantification of target miRNA without reverse transcription or PCR amplification. This direct detection strategy reduces sample preparation steps and shortens turnaround time compared with conventional RT-qPCR.

Given the stability of miRNAs and the practical advantages of the SSMC system, we explored its potential for biomarker development. In this study, we evaluated the feasibility of using SSMC for highly sensitive detection of miR-200c in lung cancer lymph node metastases, offering a faster alternative to conventional RT-qPCR.

## 2. Materials and Methods

### 2.1. Study Design and Data Source

We first assessed the correlation between SSMC and RT-qPCR results, demonstrating that SSMC is an effective tool for detecting target miRNAs. Since SSMC is a novel technology, its application to clinical specimens has rarely been reported. Secondly, the efficacy of miR-200c as a biomarker was validated using SSMC. In this study, SSMC was used to measure miR-200c concentration per needle wash. The yield of the extracted miRNA depended on the amount of specimen obtained using EBUS-TBNA. Clinically, specimens from malignant lymph nodes are known to be larger than those from benign lymph nodes. Therefore, when comparing concentrations directly, malignant lymph nodes with higher total miRNA yields may appear to have higher target miRNA concentrations. To address this problem, we used U6 as a housekeeping gene and evaluated the biomarker potential of miR-200c by relative quantification. Finally, we assessed whether miR-200c could serve as a reliable biomarker for identifying malignant lymph nodes without normalization to a housekeeping gene.

This was a collaborative study between the Chiba University Graduate School of Medicine and Olympus Medical Systems Corporation. Lymph node samples were collected at Chiba University Hospital, where all patient identification information was stored. All patient identifiers were removed before the samples were sent to the Olympus Medical Systems Corporation laboratory for SSMC analysis (Olympus Ethical Committee approved). The study was conducted in accordance with the principles of the Declaration of Helsinki.

### 2.2. Clinical Samples

Patients with lung cancer or suspected lung cancer scheduled to undergo EBUS-TBNA for the diagnosis of mediastinal lymph node metastasis were recruited between December 2017 and March 2020 at Chiba University Hospital. The ethics committee approved clinical sample collection (Approval ID: No. 1318, Chiba University Graduate School of Medicine), and written informed consent was obtained from all participants. A total of 100 samples were collected from 86 patients. Fifty-eight samples from 54 patients were used to validate the SSMC system and RT-qPCR results. An additional 42 samples from 46 patients were included, resulting in 100 overall samples for differentiating metastatic lymph nodes from benign lymph nodes. Based on these data, analyses of the area under the curve (AUC) of the receiver operating characteristic (ROC) curve were performed, and the sensitivity, specificity, and cut-off values were determined.

### 2.3. Specimen Collection

Samples were collected during EBUS-TBNA of mediastinal lymph nodes. Twenty-two-gauge biopsy needles were used for EBUS-TBNA. After obtaining the lymph node tissue for pathological examination, the needle was flushed with 1 mL saline to recover residual tissue and blood. The lavage microvolume fluid was mixed with 1 mL RNAlater (QIAGEN, Venlo, The Netherlands) and stored at −80 °C until analysis. Specimens sent to the Olympus laboratory were transported on dry ice and stored at a low temperature.

### 2.4. MicroRNA Preparation and Measurement

miRNA was extracted from the needle wash lavage fluid using the miRNAeasy kit (Qiagen, Venlo, The Netherlands) according to the manufacturer’s instructions. RT-qPCR was performed using the miRCURY LNAmiRNA Custom Probe PCR Assay (Qiagen, Venlo, The Netherlands) according to the manufacturer’s instructions. The primers used were has-miR-200c-3p (Product No. YP00204482), has-miR-200b-3p (Product No. YP00206071), and u6 snRNA (Product No. YP00203907). The measurement methods for RT-qPCR and SSMC have been reported previously [16,17] and were applied in this study.

### 2.5. Statistical Analysis

The validity of SSMC results compared with the RT-qPCR results was analyzed using simple linear regression. The Mann–Whitney U test was used to analyze the molar concentrations measured by SSMC. A *p*-value of <0.05 was considered statistically significant. ROC curve and AUC analyses were used to determine the sensitivity, specificity, and corresponding cut-off values for miR-200c. All analyses were performed using GraphPad Prism 9.3.1 for Windows (GraphPad Software, La Jolla, CA, USA; www.graphpad.com).

## 3. Results

### 3.1. Clinical Characteristics

The clinical characteristics of the patients are summarized in Table 1. All pathological results were determined using tissues acquired via EBUS-TBNA. In one case, the lung cancer type could not be identified in the sampled lymph node.

### 3.2. Correlation Between RT-qPCR and SSMC Results

First, we assessed the correlation between the RT-qPCR and SSMC results. Fifty-eight samples from 54 patients (47 metastatic and 11 benign lymph nodes) were used for validation. Linear regression analysis of miR-200c levels between SSMC and RT-qPCR showed a statistically significant positive correlation (R^2^ = 0.817, *p* < 0.0001) (Figure 2). For additional validation, we performed the same analysis for miR-200b. Linear regression analysis of miR-200b showed a statistically significant positive correlation (R^2^ = 0.759, *p* < 0.0001) (Appendix A). Because a high correlation was found between RT-qPCR and SSMC, we used the SSMC results for subsequent analyses.

### 3.3. Validation Test Between with/Without Housekeeping Gene Control

The mean miR-200c concentration was 4417.93 fM (femtomolar) in malignant lymph nodes, compared with 314.66 fM in benign lymph nodes. Mean U6 concentrations were 50,016.45 fM in malignant nodes and 61,702.25 fM in benign nodes, with no significant difference (*p* = 0.49) (Figure 3a). In the comparison of relative evaluations between malignant and benign lymph nodes using U6 as the housekeeping gene, the mean relative value of miR-200c/U6 in malignant lymph nodes was significantly higher than in benign lymph nodes (*p* < 0.0001) (Figure 3b).

The mean miR-200c concentration was significantly higher in malignant lymph nodes than in benign lymph nodes (*p* < 0.0001) (Figure 4a). In ROC analysis, miR-200c concentration showed the highest diagnostic yield as a classifier (AUC = 0.883), with a sensitivity of 89.36% and specificity of 90.91% at a cut-off value of 222.5 fM (Figure 4b).

### 3.4. Deciding the Cut-Off Value as a Biomarker of miR-200c in 100 Samples

Based on the previous results, subsequent evaluations were conducted without normalization to U6. A total of 100 samples from 86 patients were evaluated to assess miR-200c as a biomarker for identifying malignant lymph nodes. Among these, 70 were malignant, and 30 were benign lymph nodes (Table 1 and Table 2).

The mean miR-200c concentration was 3925.09 fM in malignant lymph nodes and 389.04 fM in benign lymph nodes. The mean miR-200c concentration was significantly higher in malignant than in benign lymph nodes (*p* < 0.0001) (Figure 5a). In ROC analysis, miR-200c concentration showed the highest diagnostic yield as a classifier (AUC = 0.864), with a sensitivity of 85.71% and specificity of 83.33% at a cut-off value of 222.5 fM (Figure 5b).

## 4. Discussion

SSMC has been reported as a highly sensitive method for measuring microRNAs [16,17]. However, to date, there are no reports on its use in clinical specimens. To the best of our knowledge, this is the first study to assess SSMC using clinical samples. SSMC showed a statistically significant positive correlation with RT-qPCR in the needle lavage fluid, suggesting that SSMC may be useful for evaluating clinical samples. In this study, we used a needle-washing solution, showing that SSMC has high sensitivity, even with minimal sample volumes. Conventional molecular diagnostic approaches (e.g., NGS, ddPCR) usually require complex sample preparation steps, such as nucleic acid extraction or amplification, which limit their speed. In contrast, SSMC enables direct detection of miRNAs without reverse transcription or PCR amplification. This advantage facilitates the development of a high-speed SSMC method, although further studies are warranted. Because only two miRNAs were evaluated in this study, the utility of SSMC for other miRNAs or mRNAs remains uncertain, and further studies are needed.

miRNAs are small, non-coding RNAs consisting of 19–24 nucleotides that regulate RNA silencing and post-transcriptional gene expression [18]. The miR-200 family consists of five members: miR-200a, miR-200b, miR-200c, miR-141, and miR-429, organized into two genomic clusters. Among these, miR-200c regulates the transcription factor ZEB1 and restores E-cadherin, an important cell-to-cell adhesion protein [19,20,21]. miR-200c suppresses tumor progression by inhibiting epithelial–mesenchymal transition, the initiating step of metastasis. Collectively, the miR-200 family prevents malignant transformation and cancer progression by targeting the early stages of the metastatic cascade [22,23,24]. However, elevated miR-200c expression has also been reported in advanced cancers and metastatic sites, suggesting tumor-promoting roles [25,26,27].

These contradictory findings reflect the complex biology of miR-200c. Clinically, both reduced [24] and increased [28,29,30,31] miR-200c levels in lung cancer tissue have been associated with poor prognosis, limiting its reliability as a prognostic marker. Nevertheless, several studies, including ours, indicate that miR-200c is consistently upregulated in lymph node metastases [32]. Our study showed that miR-200c can detect lymph node metastasis with high sensitivity and specificity, even in the absence of housekeeping gene controls. This finding suggests that miR-200c may complement the pathological diagnosis in the detection of lymph node metastasis. However, further studies are needed to determine whether increased miR-200c levels in lymph nodes exert a tumor-inhibitory effect.

Several limitations of this study warrant further investigation. First, this was not a prospective study. Although we obtained a cutoff value for the detection of malignant lymph nodes, its clinical utility should be validated in prospective studies. Additionally, because final diagnoses were confirmed based on pathological results from EBUS-TBNA, the possibility of false negatives (approximately 10%) [9,10] cannot be excluded. Therefore, future studies should include participants undergoing EBUS-TBNA for preoperative staging, with pathology results from surgically dissected lymph nodes. Furthermore, the requirement for confocal microscopy and specialized optical detectors currently restricts the routine use of SSMC in clinical laboratories. Nonetheless, we envision SSMC as a complementary approach to ROSE, especially in settings where rapid cytology evaluation is unavailable. While SSMC requires dedicated equipment, the reduced manpower and personnel costs compared with rapid cytology may justify its clinical value.

## 5. Conclusions

In this study, we demonstrated that SSMC is a highly sensitive method for assessing miRNAs in lymph nodes. Specifically, we showed that miR-200c detected lung cancer lymph node metastasis with high sensitivity and specificity. This evaluation method has the potential to serve as a complementary diagnostic tool and may help reduce false-negative results in clinical practice.

## Figures and Tables

**Figure 1 cancers-17-03133-f001:**
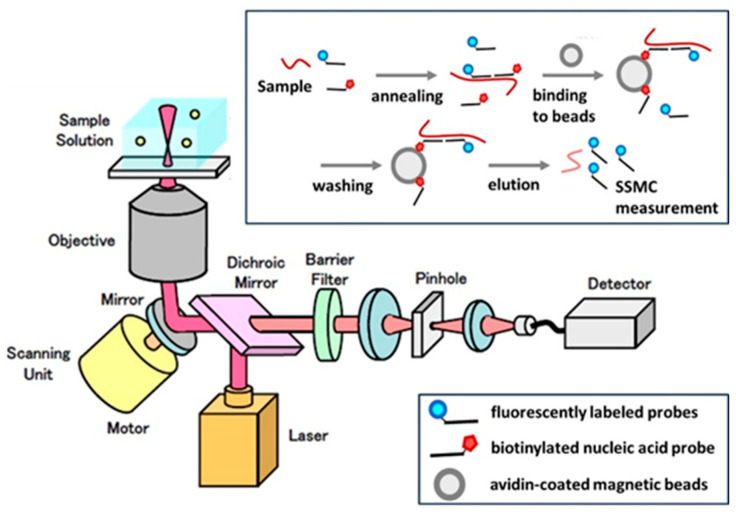
Principle of the Scanning Single-Molecule Counting (SSMC) Method.

**Figure 2 cancers-17-03133-f002:**
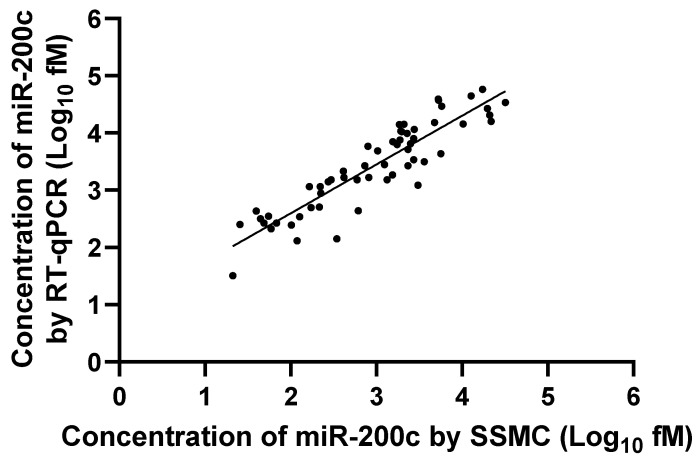
Linear regression analysis for concentrations of miR-200c between RT-qPCR and SSMC.

**Figure 3 cancers-17-03133-f003:**
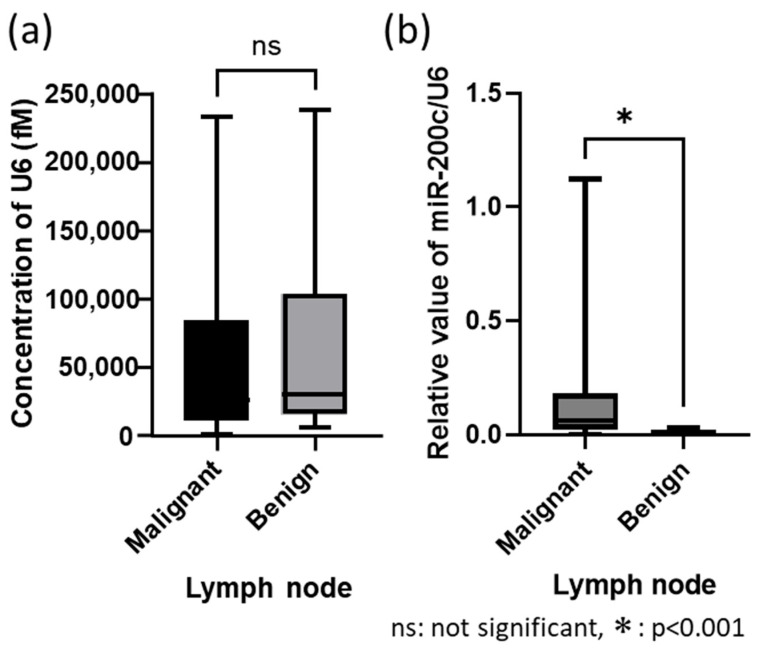
Comparison of malignant and benign lymph nodes for each microRNA. (**a**) U6 concentration. (**b**) Relative value of miR-200c/U6.

**Figure 4 cancers-17-03133-f004:**
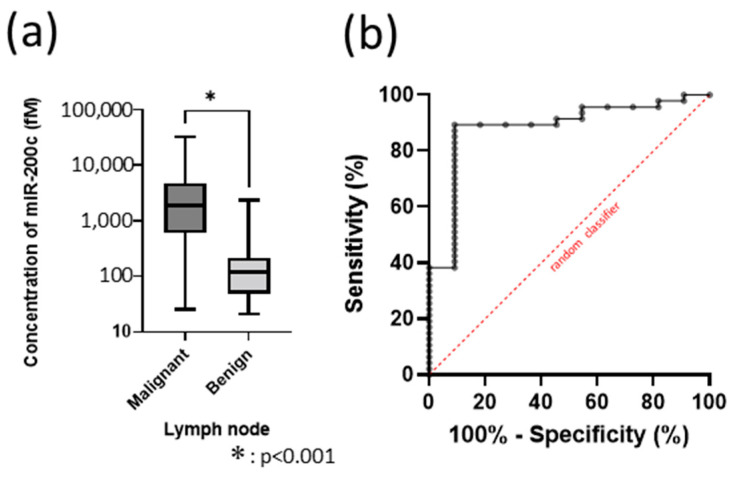
Analysis of miR-200c without U6. (**a**) miR-200c concentration in lymph nodes. (**b**) ROC curve for miR-200c measured by SSMC.

**Figure 5 cancers-17-03133-f005:**
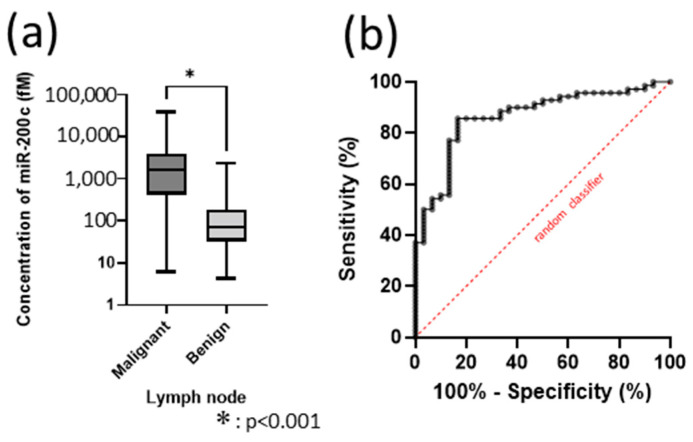
miR-200c analysis in all 100 samples. (**a**) miR-200c concentration in lymph nodes. (**b**) ROC curve for miR-200c measured by SSMC.

**Table 1 cancers-17-03133-t001:** Patient characteristics and pathological status of lymph nodes (based on pathological findings).

Patients, n	86
Male	64
Female	22
Median age (range), years	70.5 (28–86)
Lymph nodes (total), n	100
Malignant	70
Adenocarcinoma	45
Squamous cell carcinoma	17
Non-small cell carcinoma	1
Small cell carcinoma	7
Benign	30

**Table 2 cancers-17-03133-t002:** Status of lymph node used for microRNA analysis.

	Malignant, *n* = 70	Benign, *n* = 30
Purpose for EBUS-TBNA		
Staging	42	25
Confirmation for recurrence	22	5
Re-biopsy for searching for gene mutation	6	0
Treatment		
Before chemotherapy, including non-treatment	42	27
After chemotherapy	28	3

## Data Availability

The original data is unavailable due to ethical restrictions.

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
