# Peer review of "Highly Sensitive Detection of miR-200c in Metastatic Lymph Nodes Using Scanning Single-Molecule Counting"

_cancers, 2025, doi:10.3390/cancers17193133_

Round 1
Reviewer 1 Report
Comments and Suggestions for Authors
- Originality
This manuscript investigates the potential utility of miRNA analysis using SSMC for rapid nodal staging in lung cancer. The application of this method to clinical EBUS-TBNA samples is novel and has rarely been explored, supporting the study’s originality. Although clinical implementation remains preliminary, the main topic addresses an important gap in the current literature and daily practice.
- Relationship to Literature
The authors provide an appropriate study background by highlighting the gaps and clinical needs in nodal staging of lung cancer and as well as the emerging role of miRNAs as biomarkers. Relevant studies are cited, though the discussion of competing molecular diagnostic approaches could be expanded to better contextualize the contribution of SSMC.
- Methodology
The study design includes a comparative analysis with RT-qPCR and normalization to U6 as a housekeeping gene. Limitations, including the retrospective design, evaluation of a limited number of miRNAs, and lack of surgical confirmation, are acknowledged by the authors. Besides these, I would suggest clarifying whether the sensitivity and specificity estimates were internally validated and potential risk of overfitting led by determining a cut-off without external validation.
- Results
The results are clearly presented and supported with appropriate figures and tables. Diagnostic performance metrics are convincing. However, the absence of subgroup analyses regarding histological type and treatment effects limits interpretability. The abstract would be strengthened by including key numerical data to support its claims.
- Implications for Research, Practice, and Society
If validated prospectively, SSMC could serve as a rapid adjunct to cytopathology, potentially reducing the incidence of false-negative staging. However, the manuscript gives little attention to cost-effectiveness and practical aspects of integration into diagnostic workflows. A more detailed discussion of how SSMC could be implemented in clinical practice would strengthen the translational impact.
- Quality of Communication
The manuscript is well-structured and generally free of language issues with adequate flow. However, the discussion on the conflicting roles of miR-200c sounds repetitive. A more concise narrative would improve readability.
Author Response
Response to Reviewer 1
- Originality
This manuscript investigates the potential utility of miRNA analysis using SSMC for rapid nodal staging in lung cancer. The application of this method to clinical EBUS-TBNA samples is novel and has rarely been explored, supporting the study’s originality. Although clinical implementation remains preliminary, the main topic addresses an important gap in the current literature and daily practice.
Response:
We sincerely thank the reviewer for recognizing that this study addresses an important gap in the current literature and daily practice.
- Relationship to Literature
The authors provide an appropriate study background by highlighting the gaps and clinical needs in nodal staging of lung cancer and as well as the emerging role of miRNAs as biomarkers. Relevant studies are cited, though the discussion of competing molecular diagnostic approaches could be expanded to better contextualize the contribution of SSMC.
Response:
We agree with the reviewer’s suggestion to expand on the comparison with other molecular diagnostic approaches. While many approaches (e.g., NGS, ddPCR) have been reported, these typically require additional steps, such as nucleic acid extraction or amplification, which limit their rapidity. In contrast, SSMC can be applied directly to the specimen, and its technical design enables faster detection. We have added this explanation in the revised Discussion to highlight why SSMC could serve as a complementary tool to rapid on-site pathological evaluation.
Discussion (Page 6, Lines 195–197): Conventional molecular diagnostic approaches (e.g., NGS, ddPCR) usually require complex sample preparation steps, such as nucleic acid extraction or amplification, which limit their speed.
- Methodology
The study design includes a comparative analysis with RT-qPCR and normalization to U6 as a housekeeping gene. Limitations, including the retrospective design, evaluation of a limited number of miRNAs, and lack of surgical confirmation, are acknowledged by the authors. Besides these, I would suggest clarifying whether the sensitivity and specificity estimates were internally validated and potential risk of overfitting led by determining a cut-off without external validation.
Response:
As requested, we clarified that sensitivity and specificity were internally validated, and the cut-off value was determined based on the optimal point of the ROC curve (AUC = 0.88). We acknowledge the risk of overfitting when determining a cut-off without external validation, and, therefore, we clearly state in the revised manuscript that a prospective external validation study is planned.
- Results
The results are clearly presented and supported with appropriate figures and tables. Diagnostic performance metrics are convincing. However, the absence of subgroup analyses regarding histological type and treatment effects limits interpretability. The abstract would be strengthened by including key numerical data to support its claims.
Response:
In accordance with the reviewer’s suggestion, we revised the Abstract to include key numerical data (AUC=0.88) to strengthen its claims.
- Implications for Research, Practice, and Society
If validated prospectively, SSMC could serve as a rapid adjunct to cytopathology, potentially reducing the incidence of false-negative staging. However, the manuscript gives little attention to cost-effectiveness and practical aspects of integration into diagnostic workflows. A more detailed discussion of how SSMC could be implemented in clinical practice would strengthen the translational impact.
Response:
We expanded the Discussion to better describe how SSMC could be integrated into clinical practice. We envision its use as a complementary approach to ROSE and in settings where rapid cytology evaluation is unavailable. While SSMC requires specialized equipment, we argue that, when considering the manpower and personnel costs of rapid cytology, it may offer sufficient clinical value in the future.
We have added the following sentence to the Discussion section (Page 7, Lines 235–340): Furthermore, the requirement for confocal microscopy and specialized optical detectors currently restricts the routine use of SSMC in clinical laboratories. Nonetheless, we envision SSMC as a complementary approach to ROSE, especially in settings where rapid cytology evaluation is unavailable. While SSMC requires dedicated equipment, the reduced manpower and personnel costs compared with rapid cytology may justify its clinical value.
- Quality of Communication
The manuscript is well-structured and generally free of language issues with adequate flow. However, the discussion on the conflicting roles of miR-200c sounds repetitive. A more concise narrative would improve readability.
Response:
Following the reviewer’s comment, we revised the Discussion to make the section on the conflicting roles of miR-200c more concise and avoid redundancy.
Reviewer 2 Report
Comments and Suggestions for Authors
The authors emphasize a significant challenge in lung cancer diagnosis – the accurate and rapid detection of lymph node metastases. Standard methods, such as cytological evaluation of collected material using EBUS-TBNA, are effective but carry the risk of false-negative results. This study presents a single-molecule scanning microRNA (miR-200c) method that enables accurate and rapid staging of the tumor. Correlation analysis, statistical and ROC tests were performed, and biomarker sensitivity/specificity were determined.
The article is substantively reliable, well-written, and has real clinical relevance. The results add value thanks to the use of the SSMC method, implemented for the first time in clinical practice for the diagnosis of lung cancer. However, the SSMC method requires specialized laboratory equipment that is typically not available in standard clinical laboratories. SSMC utilizes confocal microscopy, which requires a dedicated optical system and highly sensitive detectors, which may limit its availability and widespread use.
The only drawback of the presented work is the lack of a schematic visualization of the SSMC method, which makes it difficult for those outside the narrow field to understand the innovative nature of the tool itself.
I suggest adding a diagram with a description explaining how the SSMC method works.
Author Response
Response to Reviewer 2
The authors emphasize a significant challenge in lung cancer diagnosis – the accurate and rapid detection of lymph node metastases. Standard methods, such as cytological evaluation of collected material using EBUS-TBNA, are effective but carry the risk of false-negative results. This study presents a single-molecule scanning microRNA (miR-200c) method that enables accurate and rapid staging of the tumor. Correlation analysis, statistical and ROC tests were performed, and biomarker sensitivity/specificity were determined.
The article is substantively reliable, well-written, and has real clinical relevance. The results add value thanks to the use of the SSMC method, implemented for the first time in clinical practice for the diagnosis of lung cancer. However, the SSMC method requires specialized laboratory equipment that is typically not available in standard clinical laboratories. SSMC utilizes confocal microscopy, which requires a dedicated optical system and highly sensitive detectors, which may limit its availability and widespread use.
The only drawback of the presented work is the lack of a schematic visualization of the SSMC method, which makes it difficult for those outside the narrow field to understand the innovative nature of the tool itself.
I suggest adding a diagram with a description explaining how the SSMC method works.
Response:
We greatly appreciate the reviewer’s positive evaluation of our manuscript as substantively reliable, well-written, and clinically relevant.
We agree with the reviewer that the requirement for confocal microscopy and specialized optical detectors currently limits the general availability of SSMC in routine clinical laboratories. We have added a statement in the Discussion to emphasize this limitation while also noting that, considering the manpower and costs required for rapid cytology, SSMC may provide sufficient clinical value in the future.
In accordance with the reviewer’s valuable suggestion, we have added a schematic diagram illustrating the principles of the SSMC method, together with a descriptive explanation, to help readers outside the field understand the innovative nature of technology.
Figure 1. Principle of the Scanning Single-Molecule Counting (SSMC) Method
The schematic illustrates the workflow of the SSMC system. Fluorescently labeled probes hybridize directly to target miRNAs in the sample solution. Using confocal laser scanning microscopy, the observation volume is illuminated, and individual fluorescent signals corresponding to single molecules are detected by highly sensitive photodetectors. Each detected fluorescent event is counted as one molecule, enabling absolute quantification of target miRNA without reverse transcription or PCR amplification. This direct detection strategy reduces sample preparation steps and shortens turnaround time compared with conventional RT-qPCR.
